# Maternal Influenza A Virus Infection Restricts Fetal and Placental Growth and Adversely Affects the Fetal Thymic Transcriptome

**DOI:** 10.3390/v12091003

**Published:** 2020-09-08

**Authors:** Hana Van Campen, Jeanette V. Bishop, Vikki M. Abrahams, Helle Bielefeldt-Ohmann, Candace K. Mathiason, Gerrit J. Bouma, Quinton A. Winger, Christie E. Mayo, Richard A. Bowen, Thomas R. Hansen

**Affiliations:** 1Animal Reproduction and Biotechnology Laboratory, Department of Biomedical Sciences, College of Veterinary Medicine and Biomedical Sciences, Colorado State University, Fort Collins, CO 80523, USA; hana.van_campen@colostate.edu (H.V.C.); jeanette.bishop@colostate.edu (J.V.B.); gerrit.bouma@colostate.edu (G.J.B.); quinton.winger@colostate.edu (Q.A.W.); richard.bowen@colostate.edu (R.A.B.); 2Department of Microbiology, Immunology and Pathology, College of Veterinary Medicine and Biomedical Sciences, Colorado State University, Fort Collins, CO 80523, USA; candace.mathiason@colostate.edu (C.K.M.); christie.mayo@colostate.edu (C.E.M.); 3Department of Obstetrics, Gynecology & Reproductive Sciences, Yale School of Medicine, Yale University, New Haven, CT 06510, USA; vikki.abrahams@yale.edu; 4Australian Infectious Diseases Research Centre, The University of Queensland, St. Lucia, QLD 4072, Australia; h.bielefeldtohmann1@uq.edu.au

**Keywords:** growth, influenza, fetus, immunity, thymus, *Mal*

## Abstract

Maternal influenza A viral infections in humans are associated with low birth weight, increased risk of pre-term birth, stillbirth and congenital defects. To examine the effect of maternal influenza virus infection on placental and fetal growth, pregnant C57BL/6 mice were inoculated intranasally with influenza A virus A/CA/07/2009 pandemic H1N1 or phosphate-buffered saline (PBS) at E3.5, E7.5 or E12.5, and the placentae and fetuses collected and weighed at E18.5. Fetal thymuses were pooled from each litter. Placentae were examined histologically, stained by immunohistochemistry (IHC) for CD34 (hematopoietic progenitor cell antigen) and vascular channels quantified. RNA from E7.5 and E12.5 placentae and E7.5 fetal thymuses was subjected to RNA sequencing and pathway analysis. Placental weights were decreased in litters inoculated with influenza at E3.5 and E7.5. Placentae from E7.5 and E12.5 inoculated litters exhibited decreased labyrinth development and the transmembrane protein 150A gene was upregulated in E7.5 placentae. Fetal weights were decreased in litters inoculated at E7.5 and E12.5 compared to controls. RNA sequencing of E7.5 thymuses indicated that 957 genes were downregulated ≥2-fold including *Mal*, which is associated with Toll-like receptor signaling and T cell differentiation. There were 28 upregulated genes. It is concluded that maternal influenza A virus infection impairs fetal thymic gene expression as well as restricting placental and fetal growth.

## 1. Introduction

Maternal viral infections are recognized to have detrimental effects on the human fetus with serious consequences for the development and viability of the child [1]. Analysis of records from the 1918 influenza pandemic discovered profound effects of maternal influenza A viral infection that persisted throughout the life of the child affecting their health, educational achievement and socioeconomic status as adults [2,3]. The impact of influenza on health parameters is also reflected in increased public entitlement spending, impacting society as well as the individual. Data collected regarding the health of pregnant women and their infants during the 2009 influenza pandemic has since confirmed the negative impact of maternal influenza infections on children [4,5]. Specifically, the link between maternal influenza infections with pre-term birth, low birth weight, congenital defects of the heart, cardiovascular disease as adults, and neurologic/behavioral abnormalities such as schizophrenia has been established [4,5,6,7,8,9,10,11,12]. Emerging data suggest that affected infants are also at risk for immune-related diseases, such as asthma or type 1 diabetes; however, the effects of maternal influenza A virus infection on the immune system of infants and children has not been extensively explored [7,13,14]. We hypothesize that in addition to negative effects on prenatal growth, maternal influenza infection during pregnancy impairs the fetal immune system such that responses to secondary infections are compromised postnatally.

The effects of maternal influenza A virus infection of the health of pregnant females and their offspring have been investigated using susceptible mouse strains with mutations in the antiviral *Mx* gene [15]. These studies have determined that maternal influenza A virus infection results in: (1) low birth weight; (2) reduced survival rates and postnatal growth rates of offspring; (3) behavioral changes and effects on brain gene expression [16,17,18,19,20,21]. Few studies have focused on the effect of maternal infections on the fetal immune system and subsequent susceptibility of the offspring to disease. One investigation found that maternal influenza A infection led to multiple changes to fetal lymphoid organs; however, effects on specific genes were not reported [22]. In a Coxsackie B virus maternal-fetal infection mouse model, significant changes in thymic T cell populations was detected by flow cytometry in day 17 gestational age fetuses, 0- and 5-day-old offspring of female mice inoculated with Coxsackie B virus on day 10 of gestation [23].

Our focus on the fetal thymus is based on previous experiments using a bovine virus diarrhea virus (BVDV) maternal infection model in cattle, in which significant changes in gene expression in the fetal thymus and spleen were found at different time points during gestation [24,25]. Moreover, thymic gene expression was influenced by the timing of maternal infection during pregnancy [25,26]. Clinical observations suggest that maternal BVDV infections have an impact on health of the offspring postnatally [27,28]. These findings suggested that maternal virus infections alter gene expression within the developing fetal thymus which may impact T cell development; and, in the long term, the adaptive immune response of offspring may be permanently damaged increasing their susceptibility to immune-mediated and infectious diseases.

To examine the effects of maternal viral infection on gene expression in the developing fetal thymus, pregnant C57BL/6 (*Mx*^−/−^) mice were inoculated intranasally with influenza A virus A/CA/07/2009 pandemic H1N1 (influenza) or phosphate-buffered saline (PBS, control) at pre-implantation (E3.5) corresponding to the period prior to day 7 in humans, peri-implantation (E7.5) corresponding to days 7 to 9 in humans, and post-implantation (E12.5), corresponding to post-day 12 pregnancy in humans [29]. Litter size, placental and fetal weights, placental morphology were examined in placenta and fetal thymic gene expression in fetuses collected on E18.5. The results indicated that in addition to decreased placental and fetal weight, maternal influenza infection during pregnancy results in decreased expression of a large number of genes in the fetal thymus and measurable changes in placental vascular development.

## 2. Materials and Methods

### 2.1. Animals

Mouse experiments were reviewed, approved, and performed in accordance with Institutional Animal Care and Use Committee at Colorado State University (IACUC Approval #16-6571A). Eight-week-old female and male C57BL/6 mice were obtained from Jackson Laboratories (Bar Harbour, ME, USA). Mice were provided with mouse chow and water ab libitum and housed in a temperature-controlled room on a 12 h light/dark cycle. Five female mice were housed in each cage and males were housed in individual cages. Females in proestrus were placed in a male’s cage in the afternoon and were examined for vaginal plugs the following morning. A gestational date of E0.5 was assigned to females with vaginal plugs.

### 2.2. Influenza Virus Inoculum Preparation and Titration

A/California/07/2009 H1N1 pandemic influenza A virus was grown in embryonated chicken eggs and aliquots of the allantoic fluid containing virus was stored at −80 °C until used as inoculum. The inoculum was titrated on Madin-Darby canine kidney (MDCK-London) cells obtained from R.A.Bowen and the dose inoculated was determined to be 6.5 × 10^4^ plaque forming units (PFU) in 25 µL. In a preliminary experiment, this intranasal dose caused transient weight loss, but no overt signs of disease or mortality in pregnant female C57BL/6 mice.

### 2.3. Experimental Design and Sample Collection

Mouse inoculations were performed in a biosafety cabinet. Pregnant females were anesthetized with 100 µg/kg ketamine + 10 µg/kg xylazine administered by intraperitoneal route. When the mice were immobile, they were held vertically and inoculated by intranasal route, i.e., the inoculum administered into both nares using a 200 µL pipettor (Rainin Mettler Toledo, Oakland, CA, USA) and sterile, filter pipet tip (Rainin 9930-384). The mice were wrapped in tissue to retain body heat, replaced in their cages and monitored until ambulatory. Pregnant females were inoculated with 25 µL of sterile PBS (sham-inoculated controls) on E3.5 (*n* = 3), E7.5 (*n* = 4) and E12.5 (*n* = 5), or with 6.5 × 10^4^ PFU influenza A virus (A/California/07/2009 H1N1) in 25 µL on E3.5 (*n* = 5), E7.5 (*n* = 5) and E12.5 (*n* = 5). Since a significant decrease in both fetal and placental weight was seen in the litters of females inoculated with influenza on E7.5 compared to controls, this time point was selected for additional maternal inoculations in Experiment 2. In Experiment 2, pregnant females were inoculated with PBS on E7.5 (*n* = 4) or with influenza A virus (*n* = 4). Female mice were weighed and observed for clinical signs of disease daily. On E18.5, females were anesthetized with 100 µg/kg ketamine + 10 µg/kg xylazine administered by intraperitoneal route, blood samples collected by cardiac puncture, and the mice were euthanized by cervical dislocation. The uterus was removed and placed into a sterile Petri dish. The individual fetuses and placentae from each litter were removed from the uterus and each of the fetuses and placentae weighed. Individual placentae were placed into microcentrifuge tubes, frozen on dry ice and stored at −80 °C for RNA extraction or fixed in 4% paraformaldehyde solution for histology. Fetal dissections were performed under a dissecting microscope (Bausch and Lomb Stereo Zoom 4, Laval, Quebec, Canada). Fine scissors were used to incise the thorax of each fetus on ventral midline, and both lobes of the thoracic thymus were removed with forceps. Because of their small size, thymuses were pooled within each litter, placed in microcentrifuge tubes, frozen on dry ice and stored at −80 °C for RNA extraction. The size of fetal thymuses also limited the extraction of protein from the same tissue. Fetal lung lobes were similarly dissected, pooled within each litter, placed in microcentrifuge tubes, frozen on dry ice and stored at −80 °C for RNA extraction. Instruments were rinsed with 70% ethanol between thymus and lung sample collection and between fetuses.

### 2.4. Histopathology

Two to four placentae from each litter were fixed in 4% paraformaldehyde, transferred to 70% ethanol after 24 h and stored until routine processed to paraffin-embedding. Four µm sections were stained with hematoxylin and eosin (H&E) and examined microscopically on a Nikon Eclipse 50i microscope (Nikon, Tokyo, Japan). Sections were also scanned using Aperio Scanner (Leica Biosystems, Wetzlar, Germany) and the width of the decidua and the labyrinth plus chorionic plate measured at the widest (middle) part of the placenta using Aperio ImageScope version 12.3.2.8013 [30]. Consecutive sections were immunolabeled for activated caspase 3 as described [31]. Microphotographs were taken of areas of the labyrinth free of artefacts at 200× magnification. To assess the fetal vascular channels, immunohistochemistry for CD34 was performed using rat anti-murine CD34 (ab8158; Abcham, Cambridge, UK) and Vector-AB anti-rat kit [32]. Using ImageJ software, 10 randomly chosen vascular channels lined by CD34 positive endothelial cells were outlined and the pixel number per luminal area recorded. Average pixel numbers from 18 control placentas and 14 placentas from influenza-inoculated dams were compared. For average count of vascular segment per area of placental labyrinth, a grid of 400 µm × 400 µm was placed over the microphotograph and all CD34 positive vascular segments within the grid were counted.

### 2.5. Hemagglutination Inhibition (HI) Assay

Infection of influenza A virus-inoculated females and the absence of infection in control females was confirmed by determining the antibody titer to A/California/07/2009 pdm H1N1 by hemagglutination inhibition (HI) [33].

### 2.6. Fetal Sex Determination

DNA was extracted from the tail of individual fetuses and the sex of the fetus determined by multiplex polymerase chain reaction (PCR) using primers for the YMT2/B locus on the Y chromosome and autosomal myogenin gene (*Myog*) [34]. The presence of sex-determining region Y (*Sry*) was indicated by the amplification product (350 bp) and the amplification of *Myog* product (250 bp) confirmed amplification of mouse DNA.

### 2.7. RNA Extraction

RNA was extracted from two to three individual placentae from each litter and from pooled fetal thymuses from sham- and influenza-inoculated female (E3.5, E7.5 and E12.5) groups using TRIzol reagent (Ambion, Carlsbad, CA, USA). RNA samples were further purified using Qiagen MiniElute columns according to the manufacturer’s instructions (Qiagen, Germantown, MD, USA). The RNA quantity and quality for quantitative real-time polymerase chain reaction (RT-qPCR) were assessed using a Nanodrop ND-1000 spectrophotometer (Thermoscientific, Waltham, MA, USA). RNA concentrations for all samples had 260/280 ratios >2.03.

### 2.8. RNA Seq of E7.5 and E12.5 Placentae and E7.5 Thymus Pools

RNA extracted from an individual placenta from each of 3 litters (E7.5 and E12.5) from Experiment 1 were selected at random for RNA sequencing (RNA seq) and analysis (provided by Ted Shade, University of Colorado Cancer Center-Genomics and Microarray Core, Aurora, CO, USA). In addition, RNA extracted from 3 control and 3 influenza-inoculated E7.5 litters pooled fetal thymuses were submitted for RNA seq. The quantity of RNA was determined using TECAN (Männedorf, Switzerland). The quality and integrity of the RNA samples was assessed using Agilent Tape Station 2200 or 4200 prior to library prep: RNA sequencing and data analysis were performed. Briefly, messenger RNA (mRNA) was purified, disrupted and a double-stranded cDNA library was synthesized using the TruSight mRNA kit (Illumina, San Diego, CA, USA). The cDNA library was sequenced using an Illumina HiSeq2500 platform and the quality of single-end (SE) reads using Real-Time Analysis (RTA) v2.7.3. FASTQ files were subjected to quality control analysis using the Rqc Bioconductor package in R. Reads were mapped using the STAR algorithm. The annotation database for alignment was built using GenomicFeatures Bioconductor package in R utilizing the University of California Santa Cruz (Santa Cruz, CA, USA) genome browser files (TXDb object) for the mouse mm10 genome. RNA-seq files were aligned to the mouse genome with the GenomeFeatures Bioconductor R package. Aligned reads were filtered based on read counts; genes with reads <2 was discarded from analysis. After filtering, 20,776 genes were analysis ready and analyzed using DESeq2 package in R. Multiple comparison adjustments were made using the Benjamini–Hochberg procedure to control the false discovery rate (FDR) method. The data have been archived in the NCBI GEO database: Accession GSE157132.

### 2.9. Pathway Analysis

The Ingenuity Pathway Analysis (IPA; Version 01-16; 2020) software package was used for the analysis of the data (Ingenuity Systems, Redwood City, CA, USA). Transcript expression was significantly different between influenza A virus-inoculated and control samples with an adjusted *p* ≤ 0.05 and fold change >1.5.

### 2.10. Influenza A Virus Quantitative Real-Time Polymerase Chain Reaction

RNA from pooled fetal lungs from each litter and RNA extracted from three individual placentae randomly selected from control and influenza A virus-inoculated E7.5 and E12.5 litters were assayed for influenza A virus matrix (M) RNA by RT-qPCR [35].

### 2.11. RT qPCR of Placentae and Thymus Pools

#### 2.11.1. Primer Design

Primers were designed using Primer Blast (NIH NCBI) and optimized for amplicons between 100 and 150 bp in length, annealing temperature of 60 °C ± 3 °C and selected for the least self-complementarity. Validation of primer sets were established by the following criteria: (1) RT-PCR of placental or thymic RNA and PCR to produce an amplicon of the predicted size, (2) DNA sequence of amplicons determined (GENEWIZ, South Plainfield, NJ, USA) to confirm identity to the target gene (NCBI Primer Blast), and (3) primer efficiency between 90 and 120% for RT-qPCR was assessed using 1:5 dilutions of cDNA from placental or pooled thymus RNA extracts. Primers for reference genes glucose-6-phosphate isomerase 1 (*Gpi1*), hypoxanthine-guanine phosphoribosyl transferase (*Hprt*) were used for placental RT-qPCR, and the reference genes *Hprt* and taldolase1 (*Taldo1*) for thymic RT-qPCR. Primer sequences and gene accession numbers are listed in Table 1.

#### 2.11.2. Target Gene Selection

Representative genes for each of the top 5 canonical pathways discovered using IPA on thymus RNA seq, were selected for validation: (1) calcium signaling pathways: actin, alpha 1, skeletal muscle (*Acta1*), (2) tight junctions: claudin 3 (*Cldn3*), (3) cellular effects and (4) hepatic fibrosis myosin light chain, phosphorylatable, fast skeletal muscle (*Mylpf*), and (5) neuroprotective: angiotensinogen (serpin peptidase inhibitor, clade A, member 8) (*Agt*). The following immune system genes were selected for further study based on their connection with thymic development: (1) myelin and lymphocyte protein (*Mal*) also known as T cell differentiation protein aka toll-interleukin 1 receptor domain containing adaptor (*Tirap*), (2) T-cell differentiation protein 2 (*Mal2*), (3) cytokine receptor like factor 1 (*Crlf1*), and (4) annexin 1 (*Anxa1*). Placental targets were: pro platelet basic protein (*Ppbp*) and transmembrane protein 150A (*Tmem150A*). See oligonucleotide primer sequences in Table 1.

#### 2.11.3. Real-Time Polymerase Chain Reaction (RT-PCR) and RT-qPCR

cDNA was synthesized using iScript RT Supermix (Bio-Rad, Hercules, CA, USA, cat#1708841) according to the manufacturer’s instructions. Duplicate samples of each pooled thymus or placental sample’s cDNA were assayed on each 384 well, clear plates (Lightcycler^®^ 480 multiwell plate 384, clear Roche Diagnostics, Indianapolis, IN, USA, cat # 05 102 430 001) and 3 separate replicate plates were assayed. Gene-specific amplification was performed on a Lightcycler 480 (Roche, Basel, Switzerland) using IQ SYBR green Supermix (Bio-Rad, cat# 170882). The reactions were performed as follows: 95 °C for 3 min, followed by 41 cycles of (95 °C × 30 s; 58 °C × 30 s; 72 °C × 15 s). Each placental target gene Ct values were normalized to the geometric mean of two references genes, glucose phosphate isomerase 1(*Gpi1*) and hypoxanthine-guanine phosphoribosyl transferase (*Hprt*). Each thymic target gene Ct values were normalized to the geometric mean of two reference genes, hypoxanthine-guanine phosphoribosyltransferase (*Hprt*) and taldolase 1 (*Taldo1*). The RT-qPCR Ct values were calculated using the comparative Ct method (Schmittgen and Livak, Nature Protocols, 2008). Ct values for 2 replicates of each gene repeated on 3 plates were averaged for each influenza and control samples. The geomean of the reference genes for the influenza samples was subtracted from the influenza average Ct for each female/litter in the influenza-inoculated group to derive the ΔCt; and the geomean of the references genes for the control samples was subtracted from the control average Ct values for each control female/litter to give the ΔCt. The 2^−ΔCt^ values for each sample were calculated.

### 2.12. Statistical Analysis

Statistical analysis of the number of fetuses per litter, fetal and placental weights, RT-qPCR data, fetal vascular lumen area and fetal capillary segments in placenta was performed in GraphPad Prism 8. The data were checked for normality using Shapiro–Wilks test. Unpaired, two-tailed Student’s *t*-tests were used to compare the fetal and placental weights, 2^−ΔCt^ values of target genes in fetal thymus for control and influenza inoculated litters. Differences between influenza-inoculated vs. controls were considered significant when *p* < 0.05. Data are presented as the mean ± standard error of the mean (SEM). The data were evaluated for normality and compared using the unpaired *t*-test (GraphPad8 GraphPad Software, San Diego, CA, USA). Correlation between E7.5 fetal and placental weights and fetal sex was analyzed using two-way analysis of variance (ANOVA, GraphPad8).

## 3. Results

### 3.1. Maternal Influenza Infection Induces Fetal Growth Restriction and Decreased Placental Weight

To determine the effect of maternal influenza infection on fetal development, we inoculated pregnant C57BL/6 mice by the intranasal route on pre-implantation (E3.5), peri-implantation (E7.5) and post-implantation (E12.5). All females were euthanized on E18.5 and individual fetuses and placentae were collected and weighed. Maternal infection was confirmed in influenza virus inoculated (E3.5, E7.5, E12.5) female mice by HI assay with positive HI titers (4 to 80) at the time of euthanasia (E18.5). All the control female mouse sera were negative for influenza virus antibodies. Clinical signs of disease were not observed, and no deaths occurred in any of the control or influenza A virus-inoculated female mice. There was no difference in the litter sizes of control E3.5, E7.5 or E12.5 and influenza virus inoculated E3.5, E7.5 or E12.5 female mice at E18.5 (Table 2). At the time of fetal sample collections (E18.5, 6 to 11 days post-maternal inoculation), influenza A viral (M) RNA was not detected in RNA extracted from three placenta per control and influenza-inoculated litters, or in RNA from pooled fetal lung samples from control and influenza-inoculated litters by RT-qPCR.

The E18.5 fetal weights from the litters of female mice inoculated with influenza at E7.5 and E12.5 were decreased compared to fetuses from control litters (Table 2); there was no difference between fetal weights of influenza-inoculated and control litters on E3.5 (*p* = 0.78) (Figure 1A). The E18.5 placental weights of litters inoculated on E3.5 and E7.5 were decreased in E3.5 and E7.5 **(**Table 2) influenza-inoculated litters compared to the controls and the placental weights of E12.5-inoculated litters were not different from controls (Table 2, Figure 1B). There was no association between the sex of the fetus by treatment and fetal weight or placental weight.

### 3.2. Placental Transcriptome and Histopathology in Influenza-Inoculated and Control Litters

To discover placental transcriptome changes caused by maternal influenza A virus infection, RNA seq was performed on an individual placenta from each of three E7.5 (Appendix A) and E12.5 influenza-inoculated (Appendix A) vs. control litters. Transcripts from four genes were increased in the RNA seq data for placentae from E7.5 maternal influenza infection compared to controls including keratin 13 (*Krt13*) (*p* = 0.001; fold change (FC) = 5.4), keratin 15 (*Krt15*) (*p* = 0.02; FC = 4.6), *Ppbp* (*p* = 0.02; FC = 4.6) and *Tmem150A* (*p* = 0.04; FC = 4.4) collected at E18.5 of gestation. Upregulation of the transcript for *Tmem150A* in E7.5 placentae (*p* = 0.0004) was confirmed by RT-qPCR in placentae (*n* = 9) from each of 4 control litters and (*n* = 13) from 5 influenza inoculated litters in Experiment 1. In contrast to the RNA seq data, RT-qPCR analysis with two different primer sets (designated Ppb4 and Ppb8 in Table 1) on RNA from E7.5 placentae showed a downregulation of *Ppbp* (*p* = 0.01, *p* = 0.04, respectively) in placentae from influenza-inoculated litters compared to controls. There were no significant differences in any transcripts between influenza-inoculated and control placentae from litters born to females inoculated on E12.5 by RNA seq.

To determine the residual effect of influenza on the placentae, two to four placentae per litter were examined histologically. Placentae from females challenged with influenza virus on E7.5 and E12.5 variably appeared with a less well-developed labyrinth, reflected morphologically both as being narrower in width compared to placentae from control mice, with a higher decidua-to-chorionic ratio and narrower vascular channels (Figure 2 and Figure 3). Frank necrosis and inflammation was only seen in the decidua of one placenta from an influenza virus-infected dam, but there was a trend for more foci of mineralization in the placental labyrinth in this cohort (1–22 foci in placenta from infected dams versus 1–11 in control dams). Immunolabeling for activated caspase 3 did not reveal any differences in apoptosis in placentae of control versus virus-challenged dams and was infrequent in both groups.

### 3.3. Maternal Influenza Infection at E7.5 Is Associated with Transcript Downregulation of 957 Genes in the Fetal Thymus

To examine the impact of maternal influenza infection on fetal thymic development, RNA-Seq was performed on fetal thymic RNA collected at E18.5 (Experiment 1). Fetal thymuses were pooled within each litter from females inoculated with influenza (*n* = 3) or PBS (*n* = 3) on E7.5. Differentially expressed genes (Bioconductor software v3.2.3) in R; 1.5-fold; *p* ≤ 0.05) were detected using the DESeq2 package. The Benjamini–Hochberg procedure was used to control the false discovery rate. Analysis (Appendix A) indicated that the fetal thymus was severely impacted by >2-fold downregulation of 957 genes and upregulation of 28 genes in influenza-inoculated dams compared to controls (*p* ≤ 0.05). IPA analysis of 1.5 fold or greater changes in gene expression is described in Figure 4.

IPA of the fetal thymus RNA seq data indicated that the first five primary canonical pathways influenced by maternal influenza infection were: (1) calcium signaling, (2) cellular effects of Sildenafil (Viagra), (3) tight junction signalling, (4) hepatic fibrosis/hepatic stellate cell activation and (5) neuroprotective role of Thimet Oligopeptidase 1 (THOP)1 in Alzheimer’s disease (Figure 4 and Appendix A). The primary differentially physiological system genes impacted by fetal influenza infection were in general organ morphology skeletal, reproductive, tissue, organismal, respiratory and embryonic development. The RNA seq data included 121 micro RNAs (miRNAs) that were activated in fetal thymuses from E7.5 influenza virus inoculated litters compared to controls (*p* < 0.05). Six activated miRNAs, miR-1237-5p, miR-3180-3p, miR-4667-5p, miR-6827-5p, miR-6873-5p and miR-7108-3p (orange symbols at the top of Appendix A) were identified by IPA analysis as upstream regulators of 40 downregulated genes (shown in green) and one upregulated gene (*Igf-1*; shown in red) associated with decreased body size and increased risk of neonatal death (see IPA pathway in Appendix A). Transcripts for 37 of the 40 downregulated genes were decreased (*p* < 0.05) in the fetal thymuses from influenza virus-inoculated litters compared to controls in the RNA seq data. In addition, four of the transcripts, *Acta1*, *Mylpf*, *Mal* and *Mal2*, were confirmed to be downregulated in E7.5 influenza virus inoculated fetal thymuses compared to controls by RT-qPCR (Figure 5). The IPA disease and function analysis of the fetal thymus RNA seq data predicted increased respiratory failure (Figure 4D); and, regulator effects analysis predicted decreased body size and increased risk of neonatal death (see Appendix A).

In the RNA-seq data from E7.5 influenza-inoculation fetal thymuses, *Mal*, a gene related to T cell differentiation/maturation was found to be significantly downregulated by 10.4-fold (*p* = 2.46 × 10^−25^) in influenza-inoculated litters compared to controls. *Mal* was identified as one of the top 10 most significantly differentially expressed genes in influenza-treated fetal thymuses (Figure 4C). Downregulation of *Mal* was confirmed by RT-qPCR in E3.5 (*p* = 0.03) and E7.5 influenza-inoculated fetal thymic RNA in Experiment 1 (*p* < 0.0001) and in Experiment 2 (*p* = 0.05) but was not different in E12.5 influenza-inoculated fetal thymic RNA (Figure 5). *Mal2*, a paralog of *Mal*, was also found to be 9-fold downregulated (*p* = 1.9 × 10^−15^) in RNA-seq data from E7.5 influenza-inoculated fetal thymuses compared to controls. By RT-qPCR, *Mal2* tended to be decreased in influenza-inoculated fetal thymic RNA in Experiment 1 (*p* = 0.1) and was decreased in Experiment 2 (*p* = 0.05) compared to controls.

## 4. Discussion

Maternal infections have long been recognized as inflicting damage on the fetus with serious consequences for the development and viability of the child [1]. Microorganisms like rubella virus, human cytomegalovirus, *Listeria monocytogenes, Toxoplasma gondii* and Zika virus infect and cross the placenta, directly infecting the fetus, leading to microencephaly and other congenital defects [36,37]. Replication of these agents incites inflammation, aberrant cell growth and/or apoptosis resulting in damage to the placenta and fetal organs and, in some cases fetal death [37,38]. Influenza A virus infections in pregnant women are strongly linked to the postnatal health of their children.

To investigate the effects of influenza A virus infection on the developing fetal immune system, we infected pregnant mice with influenza A virus and examined placental morphology, placental and fetal weight and assayed thymic gene responses by RNA seq and RT-qPCR. The significant findings are that: (1) the maternal influenza infection affects placental morphology and weight; (2) fetal weights are decreased when maternal infection occurs at pre- and peri-implantation; and (3) during the peri-implantation period of placental implantation, maternal influenza infection is associated with decreased expression of a large number of fetal thymic genes. The latter suggest that fetal immune development may be impaired which could compromise subsequent postnatal immunocompetency. The data are consistent with our hypothesis that maternal influenza infection during pregnancy leads to intrauterine growth restriction and impaired fetal thymic development. Future studies will determine if the fetal immune system is affected such that postnatal health and responses to secondary infections are compromised in the offspring.

Proposed mechanisms for the effects of maternal influenza A virus infection on the human fetus include direct infection of the placental and fetal membranes, apoptosis and placentitis [39,40,41,42,43]. In vitro culture of human-induced pluriopotent cells with influenza A, as a model of intrablastocyst infection, caused cytopathology, reduced viability and pluripotency, and autophagy [44]. Based on proteomic analysis of these stem cells, infection also caused dysregulation of many cellular pathways, including those directly associated with cellular differentiation. Influenza A viral RNA and proteins have been found in the placenta and fetus in individual cases [39,40,41,42,43,45,46,47,48]; however, documentation of direct infection is uncommon [49]. Fetal damage also occurs in the absence of direct influenza virus infection of the fetus [50,51]. Indirect mechanisms of fetal damage include hypoxia secondary to viral pneumonia in the mother, fever, changes in maternal pro-inflammatory tumor necrosis factor-α (TNF-α), interleukin (IL)-1β, IL-6 and anti-inflammatory cytokines such as IL-10 and chemokines (CXCL8, CXCL10) compared to healthy pregnant women [9,10,52,53]. Finally, maternal infections are suspected to cause epigenetic changes in the DNA of the placenta and fetus perhaps decreasing the expression of key genes necessary for healthy development and competent immune responses [54]. In this study, influenza A virus matrix gene segment RNA was not detected in placenta or fetal tissues (lung and thymus); however, these tissues were collected near term (E18.5), 6 to 15 days post-inoculation. Thus, the virus may have infected the placenta shortly after inoculation, but by the time of tissue collection, the virus may have been cleared by the maternal adaptive immune response.

Corresponding to the decreased weight of placentae from E7.5 influenza-inoculated females and to a lesser degree on E12.5, there was a reduction in the thickness of the labyrinth. This observation points to a negative influence of the infection on vasculogenesis. The vitelline vessels normally start forming around E7.5 to E8.0, and any adverse effects on this process would lead to a reduced labyrinth with consequences for nutrient and oxygen supply to the fetus and hence overall growth or adverse effect on select organs such as brain and lymphoid organs [55,56]. Influenza infection at E7.5 resulted in an increase in *Tmem150A* in placentae compared to controls that was sustained through E18.5. Increased *Tmem150A* has been described as an antiviral response in endoplasmic reticulum (ER) permeability to hepatitis C viral replication [57]. Influenza virus glycoproteins hemagglutinin (HA), neuraminidase (NA) and the matrix protein (M2) are translated into the endoplasmic reticiulum membrane. The HA induces an antiviral innate response in the ER via a stress response, which may include induction of *Tmem150A* [58]. An increase in Tmem150A may strongly reduce membrane permeability in the ER through the association of tetratricopeptide repeat domain 7A (TTC7) with phosphatidyl-inositol 4-kinase type IIIα (PI4KIIIα) compromising the placental function and contributing to decreased fetal growth [59]. It is important to note that a significant immune/inflammatory response was not observed in the placenta. The consequences of this activated antiviral response might impair membrane permeability and impact placental delivery of nutrients to the fetus, thus contributing to the fetal growth restriction. In addition, six activated miRNAs in E18.5 fetal thymuses from E7.5 influenza virus-inoculated compared to controls are upstream regulators for 40 downregulated and one upregulated gene (Appendix A). The mRNAs for 37 genes of the 40 genes were downregulated in the RNA seq data. IPA analysis indicated that these miRNAs and impacted fetal thymic gene expression were predictive of decreased body size, which is consistent with the decreased E18.5 fetal weight observed in E7.5 and E12.5 influenza virus-inoculated litters compared to control litters (see Figure 1). The suggested relationship with postnatal death will be examined in future studies clarifying effect of fetal infection with influenza on postnatal responses to secondary infections.

Most of the 957 fetal thymic genes downregulated following maternal influenza infection have functions in non-immune developmental pathways (Appendix A). Two major exceptions were the downregulation of *Mal* (10-fold) and myelin and lymphocyte protein 2 (*Mal2*) (9-fold), which were categorized as T cell differentiation/maturation proteins. Very little is known about MAL2 other than its role in transcytosis [60]. In contrast, MAL (MyD88 adaptor-like), also known as TIR domain containing adaptor protein (TIRAP), is best known for its role as a Toll-like receptor (TLR) adapter protein that bridges with the major signaling protein, MyD88 [61]. MAL is also a T cell maturation protein associated with intermediate and later stages of intra-thymic T cell differentiation and plays an essential role in T cell activation and formation of the immunological synapse between the T cell and antigen presenting cell [62,63,64,65,66]. In the activation of T cell receptors, MAL associates with lymphocyte cell-specific protein-tyrosine kinase (LCK), resulting in downstream nuclear factor kappa B subunit 1 (NFKB) activation and also plays a role in NFKB ransactivation [61,65,66,67]. Furthermore, NFKB transcriptional activity is required for thymic T cell maturation [68]. Based on this connection between MAL and NFKB, we posit that the MAL/NFKB pathway may be critical to fetal thymic development. Indeed, in our RNA-seq studies we found that NFKB mediates the transcription of 11 genes that were significantly downregulated in fetal thymic RNA (RNA seq) from dams infected with influenza including: (1) interleukin 17D (*Il17d*) a cytokine/chemokine that inhibits CD8^+^ T cell activation; (2) epidermal growth factor receptor (*Erbb)* expressed by thymic myoid cells, thymic epithelial cells and thymocytes and postulated to play a role in the induction and maintenance of tolerance; (3) transcription factor *Fosl2*, involved in the regulation of IL-1 and IL-2 activation and signaling; (4) CCAAT enhancer binding protein delta (*Cebpd*) a transcription factor involved in immune and inflammatory responses and differentiation of macrophages; and (5) E74 like erythroblast transformation Specific (ETS) transcription factor 3 (*Elf3*) important for thymic natural killer T (NKT) cell development and T cell differentiation [69,70,71,72,73,74].

The mouse fetal thymus, derived from the third pharyngeal pouch endoderm and neural crest-derived mesenchyme, is first visible at E10.5. Lymphopoietic precursor cells of bone marrow origin seed the thymus anlage at E11.5–E12.5, and by E14.5 differentiation of lymphoid cells into T cells begins. In the thymus, MAL is localized primarily to the cortex [75]. MAL has also been found in monocytes, macrophages, B cells, platelets, polarized epithelial cells, fibroblasts and endothelial cells which are constituents of the thymus [76,77,78]. At the time of influenza inoculation in our model (E7.5), the fetal thymus consists of primary endodermal and mesenchymal cells and has not yet been populated with T cell precursors; therefore, the decrease in *Mal* observed on E18.5 likely is the result of actions on gene expression 11 days earlier when endodermal cells are undergoing differentiation into thymic epithelial cells and not directly on developing T cells.

## 5. Conclusions

The large-scale inhibition of fetal thymic genes by maternal influenza infection supports further examination of the consequences of maternal infection on the fetal immune system and consequent susceptibility of the offspring to immunologic challenges. Understanding the mechanisms by which maternal influenza infection impacts pregnancy and fetal outcomes is necessary to identify targets to detect and therapeutically manage pregnancies and offspring at risk and for managing long-term postnatal impacts of the infection.

## Figures and Tables

**Figure 1 viruses-12-01003-f001:**
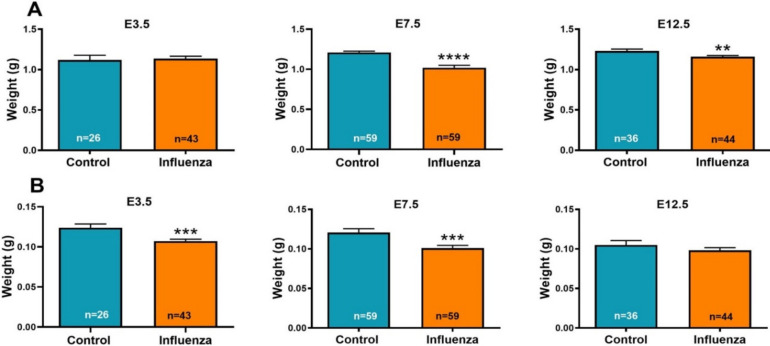
Maternal influenza reduces fetal (**A**) and placental (**B**) weights. Pregnant mice were inoculated intranasally with phosphate-buffered saline (PBS, control) or influenza at E3.5 (3 control dams; 5 influenza dams); E7.5 (8 control dams; 9 influenza dams); or E12.5 (5 control dams; 5 influenza dams). Fetuses and placentae were weighed on E18.5. Data are shown as mean ± SEM (standard error of the mean); *n* = number of fetuses. ** *p* < 0.01; *** *p* < 0.001; **** *p* < 0.0001.

**Figure 2 viruses-12-01003-f002:**
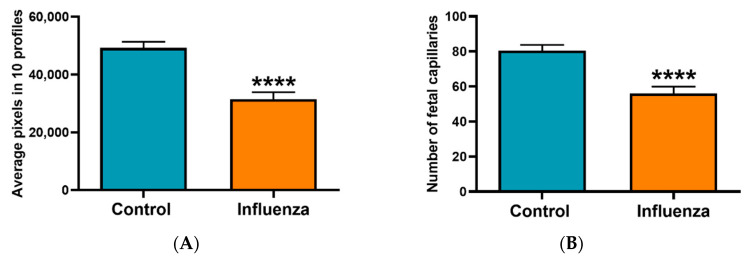
Quantification of fetal vasculature in the placental labyrinth of mice either mock-inoculated and influenza virus-inoculated on E7.5 of pregnancy and terminated on E18.5. (**A**) Average pixel numbers from 18 control placentas (blue) and 14 placentas from influenza virus-inoculated (orange) dams. Ten randomly chosen vascular channels in CD34 immunolabeled specimens from each placenta were measured and averaged. (**B**) All CD34 positive vascular segments within a 400 µm × 400 µm grid placed over the placental labyrinth were counted for 18 control placentas (blue) and 14 placentas from influenza virus-inoculated dams (orange). **** *p* < 0.0001.

**Figure 3 viruses-12-01003-f003:**
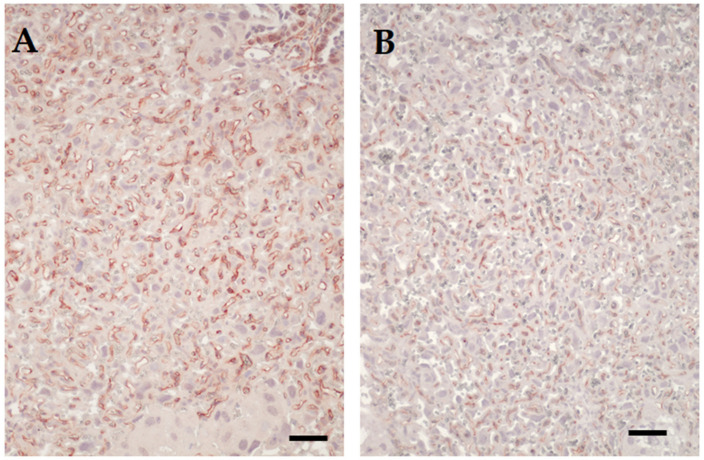
Immunohistochemical labelling of CD34 in the labyrinth of placenta. (**A**) Microphotograph of labyrinth in placenta from mock-inoculated control mouse. Fetal vessels are seen as irregular spaces lined by reddish-brown stained endothelial. Immunohistochemical labelling of CD34. Original magnification 200×. (**B**) Microphotograph of labyrinth in placenta from mouse infected on day 7.5 of pregnancy. Note reduced fetal vasculature as reflected by fewer vascular channels per area and smaller volume of individual vascular segments. Scale bars = 40 µm. Original magnification 200×.

**Figure 4 viruses-12-01003-f004:**
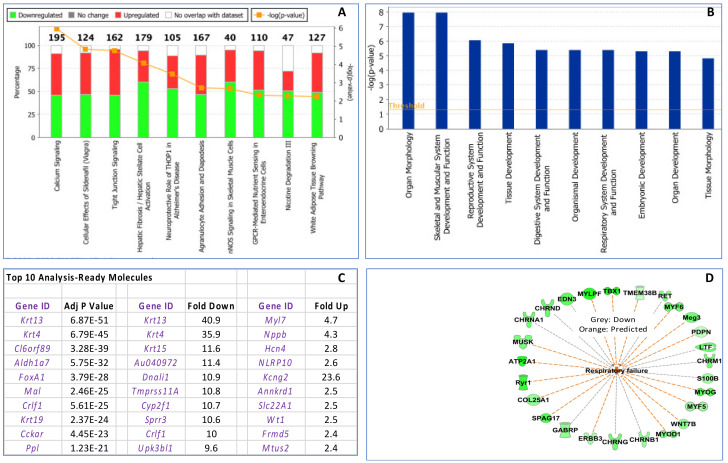
Ingenuity Pathway Analysis (IPA) analysis of genes differentially expressed 1.5-fold or greater in the E7.5 influenza-inoculated fetal thymuses compared to controls collected on day E18.5. (**A**) Top 10 canonical pathways, total number of genes in each pathway, percent of these genes downregulated and upregulated, and the −log(*p*-value). (**B**) Top 10 significant diseases and functions: physiological system: development and function pathways in E7.5 influenza inoculated compared to control fetuses collected on day E18.5. (**C**) The top 10 analysis-ready molecules based on *p* value and greatest downregulated and upregulated genes. (**D**) Disease and Function analysis identified 26 downregulated fetal thymic genes that were associated with respiratory failure in influenza inoculated compared to control fetal thymus. Also see Appendix A describing 1 upregulated and 40 downregulated fetal thymic genes in influenza inoculated thymuses associated with decreased body size and increased risk of neonatal death.

**Figure 5 viruses-12-01003-f005:**
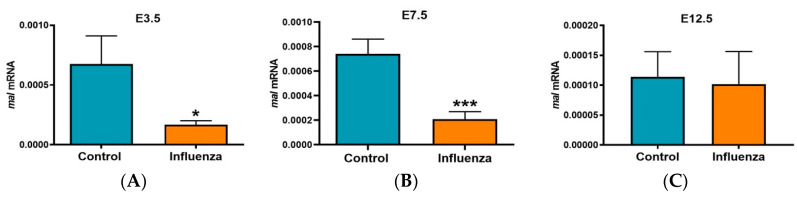
Maternal influenza infection is associated with downregulation of fetal thymic *Mal* mRNA. On E3.5, E7.5 or E12.5 pregnant mice were inoculated with PBS (control) or influenza. Subsequently, on E18.5 fetal thymuses were examined for gene expression. On E18.5 fetal thymic *Mal* mRNA was measured by RT-qPCR. Data are expressed as the relative expression (mean ± SEM). (**A**) E3.5 litters: control (*n* = 3), influenza (*n* = 5). (**B**) E7.5 litters: control (*n* = 8), influenza (*n* = 9). (**C**) E12.5 litters: control (*n* = 5), influenza (*n* = 5). * *p* < 0.05; *** *p* < 0.001.

**Table 1 viruses-12-01003-t001:** Quantitative real-time polymerase chain reaction (RT-qPCR) primer sequences. F—forward, R—reverse.

Target Gene	Accession Number	Primer Sequence
***Reference genes***		
*Gpi1*	NM_008155	F: CGAACACGGCCAAAGTGAAA
		R: AGCTGCTCGAAGTGGTCAAA
*Hprt*	NM_013556	F: AGTCCCAGCGTCGTGATTAG
		R: TCTCGAGCAAGTCTTTCAGTCC
*Taldo1*	NM_011528	F: TTGGCTGCACAACGAAGACC
		R: ATTCGTTCCGTGAGCATCCG
***Thymus Canonical Pathway Genes***		
*Acta1*	NM_001272041	F: CGCCAGCCTCTGAAACTAGA
Calcium (Ca^2+^) signaling		R: ACGATGGATGGGAACACAGC
*Cldn3*	NM_009902	F: GCAAGGACTACGTCTGAGGG
Tight junction		R: ACTGTGTGTCGTCTGTCACC
*Mylpf*	NM_016754	F: ACCACGGTATGTTAAGGGCTG
Hepatic fibrosis		R: CCTTCTTGGGTGCCATGTCTTA
*Agt*	NM_007428	F: GTTGGCGCTGAAGGATACACA
Neuroprotective		R: GACCCAGGTCAAGATGCAGAA
***Thymus Immune Systems Genes***		
*Mal*	NM_001171187	F: GTGAGTTTGATGCAGCCTACC
		R: CCACTGCGGCGATGTTTTC
*Mal2*	NM_178920	F: GGACGTACTCCGGAGCTTTC
		R: AGCTGTCACCGACACAAACA
***Placenta Genes***		
*Tmem150A*	NM_144916.3	F: TGAACAAGGGGGCCCTAAGA
		R: AGATGAGGGCCACCATAACAG
*Ppbp4*	NM_023785.3	F: TGCTGATGTGGAAGTGATAGCC
		R: GAAGCAGCTGGTCAGTAACCT
*Ppbp8*	NM_023785.3	F: ACAGCTGGAAAATCTGATGGCA
		R: CTCCTGGCCTGTACACATTCA
***A/California/07/2009(H1N1)***		
*Matrix*	NC_026431.1	F: AGATGAGTCTTCTAACCGAGGTCG
		R: TGCAAAGACACTTTCCAGTCTCTG
***Sexing genes***		
*Sry*	NM_011564.1	F: CTGGAGCTCTACAGTGATGA
		R: CAGTTACCAATCAACACATCAC
*Myog* (control)	M95800.1	F: TTACGTCCATCGTGGACAGCAT
		R: TGGGCTGGGTGTTAGTCTTAT

**Table 2 viruses-12-01003-t002:** Unpaired Student’s t-test comparing control and influenza A virus litter characteristics of females inoculated on E3.5, E7.5 and E12.5.

Day of Inoculation	E3.5			E7.5			E12.5		
	Control	Influenza	*p*	Control	Influenza	*p*	Control	Influenza	*p*
Number of litters	3	5		8	9		5	5	
Total number of fetuses	26	43		59	59		36	44	
Average number of fetuses per litter ± standard error of the mean (SEM)	8.7 ± 0.7	8.6 ± 0.5	0.9456	7.38 ± 0.75	6.56 ± 0.78	0.4768	7.2 ± 0.6	8.8 ± 0.8	0.1622
Average fetal weight per litter (g) ± SEM	1.16 ± 0.21	1.12 ± 0.07	0.8317	1.20 ± 0.02	1.03 ± 0.09	0.0674	1.25 ± 0.06	1.17 ± 0.04	0.2862
Average fetal weight (g) ± SEM	1.12 ± 0.06	1.14 ± 0.01	0.7844	1.21 ± 0.02	1.02 ± 0.03	<0.0001	1.23 ± 0.02	1.16 ± 0.01	0.0067
Average placental weight per litter (g) ± SEM	0.13 ± 0.01	0.11 ± 0.002	0.0364	0.12 ± 0.01	0.10 ± 0.01	0.1432	0.11 ± 0.02	0.10 ± 0.01	0.4901
Average placental weight (g) ± SEM	0.12 ± 0.005	0.11 ± 0.002	0.0008	0.121 ± 0.01	0.101 ± 0.003	0.0013	0.105 ± 0.01	0.098 ± 0.003	0.2740

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
