# Peer review of "Maternal Influenza A Virus Infection Restricts Fetal and Placental Growth and Adversely Affects the Fetal Thymic Transcriptome"

_viruses, 2020, doi:10.3390/v12091003_

Round 1

Reviewer 1 Report

Abstract

The authors should work on clarity and structure in the Abstract.

For example, in the first sentence, it is not clear whether the statement “maternal influenza A infections are associated …” refers to mice or humans.

Line 24: “the developing fetal immune system” is a too broad statement; towards immunity, the authors only tested thymus transcriptome.

At the end of the Abstract, it is written that intrauterine growth restriction due to maternal influenza is author's hypothesis. However, the first sentence of the abstract states the same as the already proved fact.

Introduction

The language and structure of the Introduction requires significant improvement.

For example, Line 37, “These maternal influenza A pregnancies are associated with … ”, can be rephrased much better. There are many confusing sentences.

There is no chronology and the strait introductory line. First, potential sequelae in offspring is described, then data from historical epidemics, then again details on sequelae, and finally RSV infection. It isn't easy to follow.

The important introductory statement at Line 57 – “Despite this, and the correlation …” and sentence at Line 63 – “One study found …” are contradictory. First, the authors state that immunopathology in affected fetuses and neonates was not studied. Later, they cite the paper where changes in fetal lymphoid organs were studied.

Lines 68-69: How E3.5 and E12.5 correspond to human pregnancy is not described.

Materials and Methods

Why PBS was used for mock-inoculation in mice, but not fluids from mock-inoculated eggs?

Viruses journal requires a rigorous and detailed description of MM. Please, describe technical details on how mice were inoculated.

Tissue sampling is not well described. The authors mentioned fetal lungs in Results but did not describe lung sampling. It is not clear how thymus was identified; photos are not provided. It should be difficult to identify thymus without a stereoscope. Please, justify whether thymus was accurately sampled; for example, whether thymus was not contaminated with adjacent tissues.

A histopathology section is confusing. The authors start with a description of HE staining, then describe the counting of CD34 cells. Immunohistochemistry procedures for CD34 staining are not provided at all. Information on the phenotype, function, and importance of CD34 cells is not provided in the manuscript.

It is difficult to understand how many samples were included for RNA-seq analysis. The table with individual fetuses and transparent information on the number of tested samples will be helpful. If I understood correctly provided information (Line 129), only two-three placental samples per litter were subjected for RNA extraction for virus-specific PCR and RNA-seq. This is not a sufficient number. In general, selective testing of fetuses in animal pregnancy models with multiple fetuses is not methodologically correct. Maternal infection may affect individual placentae and fetuses differently, even within the same litter. Such a selective approach promotes low methodological standards for the field.

The quality and quantity of RNA for RNA-seq was measured by Nanodrop. This is not an acceptable method for illumine kits. RNA integrity should be measured by an Agilent bioanalyzer (or acceptable alternative methods). RNA-seq data may be biased.

Statistics: BVDV?

Results

Thymus samples were pooled that makes sex-based analysis impossible. It is well-known that the sex of fetuses and offspring may affect responses to maternal or in utero infections. Thus, the authors did not account for this critical host factor during their morphological, anatomical, and molecular data analysis.

The same comment for placental samples, no sex-based analysis is described.

Sex-based statistics for the correlation between sex and weight are not described in MM.

The authors selected E7.5 for more rigorous molecular analysis based on a reduction in placental and fetal weights. However, molecular sequelae may persist without phenotypical changes. More individual samples from all groups—E3.4, E7.5, and E12.5—should be tested and analyzed.

A supplementary Figure with RNA-seq data is not readable.

Discussion

Line 345-346: Too strong statement. The authors did not establish the model. Influenza infection of pregnant mice has been previously described.

Line 351-354: The hypothesis is stated for the first time here. It seems too late.

Line 347-348: This is the overstatement of findings. The data do not show the dependency on placental development. The study is not functional but more correlative and descriptive. However, for better correlation, more samples and pregnancy stages should be tested.

Author Response

Abstract

The authors should work on clarity and structure in the Abstract.

Response:  The abstract has been restructured. 

For example, in the first sentence, it is not clear whether the statement “maternal influenza A infections are associated …” refers to mice or humans.

Response:  line 22, “in humans” has been inserted for clarification.

Line 24: “the developing fetal immune system” is a too broad statement; towards immunity, the authors only tested thymus transcriptome.

Response: line 24, “thymus” has been substituted for “immune system”.

At the end of the Abstract, it is written that intrauterine growth restriction due to maternal influenza is author's hypothesis. However, the first sentence of the abstract states the same as the already proved fact.

Response: line 30-31, “causes intrauterine growth restriction” has been deleted. 

Introduction

The language and structure of the Introduction requires significant improvement.

For example, Line 37, “These maternal influenza A pregnancies are associated with … ”, can be rephrased much better. There are many confusing sentences.

Response: line 37, has been reworded.

There is no chronology and the strait introductory line. First, potential sequelae in offspring is described, then data from historical epidemics, then again details on sequelae, and finally RSV infection. It isn't easy to follow.

Response: The Introduction has been rewritten.

The important introductory statement at Line 57 – “Despite this, and the correlation …” and sentence at Line 63 – “One study found …” are contradictory. First, the authors state that immunopathology in affected fetuses and neonates was not studied. Later, they cite the paper where changes in fetal lymphoid organs were studied.

Response:  The introduction has been rewritten and these statements omitted.

Lines 68-69: How E3.5 and E12.5 correspond to human pregnancy is not described.

Response:  The statement has been modified to include approximate correlation between the stage of placentation in the mouse with human pregnancy.

 “To examine the effects of maternal viral infection on gene expression in the developing fetal thymus, pregnant C57BL/6 (Mx-/-) mice were inoculated intranasally with influenza A virus A/CA/07/2009 pandemic H1N1 (influenza) or PBS (control) at pre-implantation (E3.5) corresponding to the period prior to day 7 in humans , peri-implantation (E7.5) corresponding to days 7 to 9 in humans  , and post-implantation (E12.5), corresponding to post-day 12 pregnancy in humans [29].

Materials and Methods

Why PBS was used for mock-inoculation in mice, but not fluids from mock-inoculated eggs?

Response: We used PBS for the mock-inoculation because our main concern was that restraint, anesthesia and inhalation of the inoculum would cause a stress response, i.e., corticosteroid release, in the female mice that might confound any effects on gene expression in the fetal thymus. We selected PBS based on other influenza A virus infection of pregnant mouse models in which control substance-inoculated pregnant mice were included in the experimental design (e.g., Williams and Mackenzie, 1977; Aronsson F., et al., 2002; Shi et al, 2003). We had not considered using allantoic fluids from mock-inoculated eggs at the time of these experiments but will include controls inoculated allantoic fluids from mock-inoculated eggs in future studies.

Viruses journal requires a rigorous and detailed description of MM. Please, describe technical details on how mice were inoculated.

Response:  lines 91-95, details of inoculation were added.

“Mouse inoculations were performed in a biosafety cabinet. Pregnant females were anesthetized with 100 ug/kg ketamine + 10 ug/kg xylazine administered by intraperitoneal route.  When the mice were immobile, they were held vertically and inoculated by intranasal route, i.e., the inoculum administered into both nares using a 200 ul pipettor (Rainin Mettler Toledo) and sterile, filter pipet tip (Rainin 9930-384). The mice were wrapped in tissue to retain body heat, replaced in their cages and monitored until ambulatory.”

Tissue sampling is not well described. The authors mentioned fetal lungs in Results but did not describe lung sampling. It is not clear how thymus was identified; photos are not provided. It should be difficult to identify thymus without a stereoscope. Please, justify whether thymus was accurately sampled; for example, whether thymus was not contaminated with adjacent tissues.

Response:  A dissecting microscope (Bausch and Lomb Stereo Zoom 4) was used to identify and aid in the dissection of the thymuses.  Honestly, it did not occur to us to obtain photos of the fetal thymuses as collection of fetal thymuses is routinely performed in our lab, and images of fetal mouse thymuses are readily available on the internet and in textbooks for comparison.  In preparation for fetal thymus dissections, the first author made squash preps of fetal thymuses and the slides were stained with Diff-Quik. Identity of the thymus made based not just on anatomical location, but on the abundant population of small, round mononuclear cells with dense chromatin and scant cytoplasm that morphologically appeared to be lymphocytes in the squash preps.  Dissecting the thymuses free of other tissues was not difficult as there are no major organs adjacent to the thymuses that could inadvertently contaminate the thymuses and the thymuses are not strongly adherent to adjacent structures.  Small sharp forceps and scissors used for dissection of the thymuses were cleansed with 70% ethanol in between each fetus.  The lung sampling was performed similarly and is briefly described. More details have been included in this section to clarify the technique.

“Fetal dissections were performed under a dissecting microscope (Bausch and Lomb Stereo Zoom 4). Fine scissors were used to incise the thorax of each fetus on ventral midline, and both lobes of the thoracic thymus were removed with forceps.  Thymuses were pooled within each litter, placed in microcentrifuge tubes and frozen on dry ice and stored at -80oC for RNA extraction. Fetal lung lobes were similarly dissected, pooled within each litter, placed in microcentrifuge tubes and frozen.  Instruments were rinsed with 70% ethanol between thymus and lung sample collection and between fetuses.”

A histopathology section is confusing. The authors start with a description of HE staining, then describe the counting of CD34 cells. Immunohistochemistry procedures for CD34 staining are not provided at all. Information on the phenotype, function, and importance of CD34 cells is not provided in the manuscript.

Response:  Lines, 115-129. This section has been edited and the procedure for the CD34 is referenced.

It is difficult to understand how many samples were included for RNA-seq analysis. The table with individual fetuses and transparent information on the number of tested samples will be helpful. If I understood correctly provided information (Line 129), only two-three placental samples per litter were subjected for RNA extraction for virus-specific PCR and RNA-seq. This is not a sufficient number. In general, selective testing of fetuses in animal pregnancy models with multiple fetuses is not methodologically correct. Maternal infection may affect individual placentae and fetuses differently, even within the same litter. Such a selective approach promotes low methodological standards for the field.

Response: Placentae from each litter were divided to provide at least 2 for histopathology and IHC, and at least 2 for RNA extraction (depending on litter size). A few were used for an exploratory flow cytometry assay (data not included in this manuscript). RNA from 3 individual placentae randomly selected from control E7.5 and E12.5 and 3 individual placentae from influenza-inoculated litters were submitted for RNA seq and comparison due to budget constraints.  In retrospect, we might have submitted RNA pooled from all available placentae (i.e., not fixed for histopathology) from each of 3 litters for RNA seq (however, the sexes would have been mixed – see reviewer’s concern below).  All RNA samples extracted from individual placentae not used for histopathology were included in and assayed individually in the RTqPCR assays. This section has been edited to try to make the number of samples clearer.

The number and sex of fetuses/thymuses used for RNA seq is shown in the Table below.  This table can be inserted in the text or included in supplementary materials.  Please let us know which you would prefer.

Litter/female ID

Control

Litter/female ID

Influenza

female

male

female

male

Bnon

2

2

A1R

3

7

B2L

2

6

B2R

3

2

C non

5

5

D2R

6

0

C2R

3

5

F1L

3

1

M1R

3

5

H2R

3

4

M1L

1

2

Mnon

2

2

M2R

7

2

M2L

5

3

O2R

4

4

O1L

4

6

O1R

2

3

Total

27

31

31

28

The fetus ID#, number and sex of placentae used for RNA seq is contained in the Table below.  This information can be inserted in the text or included in the supplementary materials.  Please let us know which you would prefer.

E7.5 control

E7.5 influenza

E12.5 control

E12.5 influenza

ID#

sex

ID#

sex

ID#

sex

ID#

sex

C2R1

female

B2R3

male

B1L4

female

E1L3

female

Cnon2

male

D2R3

female

D1L3

male

E2L3

female

B2L1

male

A1R3

male

D2L3

female

E2R3

female

The quality and quantity of RNA for RNA-seq was measured by Nanodrop. This is not an acceptable method for illumine kits. RNA integrity should be measured by an Agilent bioanalyzer (or acceptable alternative methods). RNA-seq data may be biased.

Response: The RNA was measured by Nanodrop before embarking on qRTPCR.  The sentence has been edited for clarification.

“The RNA quantity and quality for qRTPCR were assessed using a Nanodrop ND-1000 spectrophotometer (Thermoscientific). RNA concentrations for all samples had 260/280 ratios >2.03.”

RNA integrity for the RNA seq was measured by:  The paragraph is edited for clarification.

“The quantity of RNA was determined using TECAN.  The quality and integrity of the RNA samples was assessed using Agilent Tape Station 2200 or 4200 prior to library prep: RNA sequencing and data analysis were performed.”

Statistics: BVDV?

Response:  line 220, Corrected to “influenza-inoculated”

Results

Thymus samples were pooled which makes sex-based analysis impossible. It is well-known that the sex of fetuses and offspring may affect responses to maternal or in utero infections. Thus, the authors did not account for this critical host factor during their morphological, anatomical, and molecular data analysis.

Response:  Fetal thymus samples were very tiny (approximately 1 mm in diameter); therefore, they were pooled within each litter to obtain sufficient RNA for RT-qPCR and RNA seq.  It would be difficult to extract sufficient RNA (and protein) from thymuses from individual fetuses to perform RT-qPCR for multiple genes and analyze them according to fetal sex. The sex of each fetus was determined by PCR; the number of male and female fetuses in each control and influenza-inoculated E7.5 litter are shown in a table below. The number of female or male fetuses in the control and influenza-inoculated litters is not significantly different (by T test).  We can include Table 1 (same as above) in the text or as supplementary material.

 Table 1.  Sex of fetuses in each thymus pool.

Litter/female ID

Control

Litter/female ID

Influenza

female

male

female

male

Bnon

2

2

A1R

3

7

B2L

2

6

B2R

3

2

C non

5

5

D2R

6

0

C2R

3

5

F1L

3

1

M1R

3

5

H2R

3

4

M1L

1

2

Mnon

2

2

M2R

7

2

M2L

5

3

O2R

4

4

O1L

4

6

O1R

2

3

Total

27

31

31

28

The same comment for placental samples, no sex-based analysis is described.

Response: We can include Table 2 (same as in previous response) in the text or as supplementary material.

Table 2. Sex of placentae from which RNA was extracted and submitted for RNA seq.

E7.5 control

E7.5 influenza

E12.5 control

E12.5 influenza

ID#

sex

ID#

sex

ID#

sex

ID#

sex

C2R1

female

B2R3

male

B1L4

female

E1L3

female

Cnon2

male

D2R3

female

D1L3

male

E2L3

female

B2L1

male

A1R3

male

D2L3

female

E2R3

female

The same number of placentae from female fetuses (2) and male fetuses (1) were included from the E7.5 control and influenza-inoculated litters. The placentae from E12.5 control fetuses included 2 female fetuses and 1 male fetus; all of the placentae in the E12.5 influenza-inoculated group were females.  In spite of the unequal number of male and female placentae in the E12.5 groups, the RNA seq indicated that there were no differentially expressed genes in the E12.5 inoculated placentae. In future experiments, we hope to generate more litters that will give a larger number of placentae of both sexes and allow sex-based statistics with respect to placental morphology.

Sex-based statistics for the correlation between sex and weight are not described in MM.

Response:  The statistics (two-way ANOVA, Graphpad 8) for correlation between sex and weight have been added to the section on statistics. 

“Correlation between E7.5 fetal and placental weights and fetal sex was analyzed using two-way ANOVA (GraphPad8).”

The authors selected E7.5 for more rigorous molecular analysis based on a reduction in placental and fetal weights. However, molecular sequelae may persist without phenotypical changes. More individual samples from all groups—E3.4, E7.5, and E12.5—should be tested and analyzed.

Response:  With additional funding, we hope to perform additional experiments in the future to generate more individual placental samples particularly from E3.5 and E12.5 time points.

A supplementary Figure with RNA-seq data is not readable.

Response: We would like to retain this figure as a high quality tiff file in the Supplementary material so that other investigators may perhaps find genes of interest for further study within a global context.  As a tiff file, one can use the zoom feature to enlarge text which remains sharp and legible as it is enlarged. If this is not acceptable, then we can delete this figure.

Discussion

Line 345-346: Too strong statement. The authors did not establish the model. Influenza infection of pregnant mice has been previously described.

Response:  We agree and have edited this sentence to be more accurate. 

“To investigate the effects of maternal influenza infection on the developing fetal immune system, we infected pregnant mice with influenza A virus and examined thymic gene responses by RNA seq and RT-qPCR.”

Line 351-354: The hypothesis is stated for the first time here. It seems too late.

Response: We agree and have introduced the hypothesis to the reader in the Introduction (Lines 56-58).

“We hypothesize that in addition to negative effects on prenatal growth, maternal influenza infection during pregnancy impairs the fetal immune system such that responses to secondary infections are compromised postnatally.”

Line 347-348: This is the overstatement of findings. The data do not show the dependency on placental development. The study is not functional but more correlative and descriptive. However, for better correlation, more samples and pregnancy stages should be tested.

Response:  We have edited this paragraph to read:

“The significant findings are that: 1) the maternal influenza infection affects placental morphology and weight, 2) fetal weights are decreased when maternal infection occurs at pre- and peri-implantation, and 3) during the peri-implantation period of placental implantation, maternal influenza infection is associated with decreased expression of a large number of fetal thymic genes.”

Reviewer 2 Report

This is a generally solid study that examines changes induced by IAV A/Ca/07/2009 in the maternal murine transcriptome, in the hopes of identifying molecular mechanisms that could explain some of the observed human impairments following maternal infection during pregnancy. This study will be of interest to a broad readership, but a few improvements and modifications would make it stronger.

Major points

  1. Transcription studies are worthwhile, but given the general lack of strong correlation between gene expression and protein expression, it would have been worthwhile to extend this study by examining some of the proteins involved. What happens to the MAL protein? Does its gene downregulation also translate to protein down-regulation?
  2. Some studies that examine IAV-induced proteomic alterations in the context of suspected infant impairments have been published and some comparison of these authors' results with these previous studies would strengthen this study. This also applies to the possibility of gestational age relationships to disease severity briefly mentioned at line 347
  3. When genes are discussed they should be italicised. This would reduce potential confusion as to whether proteins or genes are being discussed. This occurs in many places, including, but not limited to, lines 321, 323 and 375.

Minor points

4. First 2 paragraphs of Results (lines 224 and 232). I believe authors mean Table 2, not Table 1?

5. There is an unnecessary paragraph insert at line 261.

6. Please insert scale bars onto Fig. 3. Indicating "original magnification 200X" at end of legend does not provide sufficient information about final magnification.

7. Fig 4 is lacking the heatmap and rlog expression key.

8. It is not clear at beginning of paragraph starting at line 355 whether mice or humans are being discussed. Please clarify.

9. Are "Investigators, A.I." really authors in references 3 and 5?

Author Response

Reviewer 2

This is a generally solid study that examines changes induced by IAV A/Ca/07/2009 in the maternal murine transcriptome, in the hopes of identifying molecular mechanisms that could explain some of the observed human impairments following maternal infection during pregnancy. This study will be of interest to a broad readership, but a few improvements and modifications would make it stronger.

Major points

  1. Transcription studies are worthwhile, but given the general lack of strong correlation between gene expression and protein expression, it would have been worthwhile to extend this study by examining some of the proteins involved. What happens to the MAL protein? Does its gene downregulation also translate to protein down-regulation?

Response:  We understand the limitations of interpreting transcriptome data vs protein expression.  Insufficient amounts of protein for western blot were obtained in these experiments due to the small size of the fetal thymuses.  Future plans include repeating the maternal inoculations and fetal dissections to extract enough thymic protein to examine MAL protein expression.

  1. Some studies that examine IAV-induced proteomic alterations in the context of suspected infant impairments have been published and some comparison of these authors' results with these previous studies would strengthen this study. This also applies to the possibility of gestational age relationships to disease severity briefly mentioned at line 347.

Response:  We thank the reviewer for bringing these studies to our attention and are very interested in learning more about IAV and proteomics.  However, we were unable to find these references searching with PubMed and Google Scholar.  If the reviewer could provide some of the authors names, we would gladly incorporate their findings and revise the manuscript accordingly. 

The sentence Line 347 has been revised per Reviewer 1.

  1. When genes are discussed they should be italicised. This would reduce potential confusion as to whether proteins or genes are being discussed. This occurs in many places, including, but not limited to, lines 321, 323 and 375.

Response:  The genes have been italicized. 

Minor points

  1. First 2 paragraphs of Results (lines 224 and 232). I believe authors mean Table 2, not Table 1?

Response:  Corrected.

  1. There is an unnecessary paragraph insert at line 261.

Response:  Corrected.

  1. Please insert scale bars onto Fig. 3. Indicating "original magnification 200X" at end of legend does not provide sufficient information about final magnification.

Response:  The scale (Scale bars = 40 μm) has been added to Figure 3 legend.

  1. Fig 4 is lacking the heatmap and rlog expression key.

Response:  The heatmap was deleted because labels were impossible to read in normal text format within the text of the manuscript.  We did not include it in supplementary files either because it is a partial analysis provided from the first person who completed preliminary bioiformatic analysis.  Thank you for bringing this to our attention.  We did replace this heat map to some degree with newer IPA analysis of canonical pathways that now shows total number of genes in the pathway as well as the percent of these genes that were either up or downregulated at 1.5 fold or greater and the sliding scale of adjusted P values.  Please let us know if this substitution is acceptable.

  1. It is not clear at beginning of paragraph starting at line 355 whether mice or humans are being discussed. Please clarify.

Response:  This paragraph refers to findings in human pregnancies; “human” has been inserted in the first sentence for clarification.

“Proposed mechanisms for the effects of maternal influenza A virus infection on the human fetus include direct infection of the placental and fetal membranes, apoptosis and placentitis [56-60].”

  1. Are "Investigators, A.I." really authors in references 3 and 5?

Response:  These references were removed.

Reviewer 3 Report

The objective of this study was to determine the effect of an influenza A virus 2009 pandemic H1N1 virus on pregnancy outcomes in a susceptible pregnant mouse model (Mx-\-) at different times in gestation. And innovative aspect of the article is a focus on the effect of influenza A virus infection on fetal thymic development.  The identification of Mal and Mal2 as significantly downregulated genes in the fetal thymus is a key finding of the manuscript and has implications for the developmental origins of adult immune health. This is a very important aspect of the article and increases the impact.

Specific Comments:

  1. The methods used in the study are appropriate and this reviewer appreciates the quantitation of the placental labyrinth vascular network. It would be helpful if the authors could comment on how the inoculum that they used compares to the inoculum used in other published manuscripts of influenza virus infections and mice in pregnant mice.
  2. The specific rationale for studying transcriptomics of the fetal thymus should be better developed in the introduction.
  3. A few other manuscripts have evaluated the effect of influenza A virus infection on pregnancy/fetal outcomes in the mouse model some of whom were cited (first authors Littauer, Chan) and some were not (Marcelin). Please make sure that citations of other influenza studies in pregnant mice are more comprehensive as there haven’t been so many studies.
  4. The number of validated genes from the fetal thymus appear to be relatively few. In the fetal thymus, they validated only 3 genes. It would be ideal for the authors to validate a few additional genes in the fetal thymus as this is the main focus and innovation of the study.
  5. Presentation of the transcriptomics analyses could be more comprehensive. The heat map shown in figure 4 that focuses only on the top 50 differentially regulated genes is not particularly helpful.  It would be more helpful to see a heat map based on canonical pathways to see how changes in gene expression between influenza cases and controls varies according to inoculation time in gestation.  It would also be nice to see heat maps that reflect specific gene sets of interest from the Broad Institute that might better reflect how the cases and controls defer for specific pathways of interest.

Author Response

Reviewer 3

The objective of this study was to determine the effect of an influenza A virus 2009 pandemic H1N1 virus on pregnancy outcomes in a susceptible pregnant mouse model (Mx-\-) at different times in gestation. And innovative aspect of the article is a focus on the effect of influenza A virus infection on fetal thymic development.  The identification of Mal and Mal2 as significantly downregulated genes in the fetal thymus is a key finding of the manuscript and has implications for the developmental origins of adult immune health. This is a very important aspect of the article and increases the impact.

 Specific Comments:

  1. The methods used in the study are appropriate and this reviewer appreciates the quantitation of the placental labyrinth vascular network. It would be helpful if the authors could comment on how the inoculum that they used compares to the inoculum used in other published manuscripts of influenza virus infections and mice in pregnant mice.

Response:  We used 6.5 x 104 PFU in 25 ul based on a preliminary titration experiment in pregnant C57BL6 mice.  This inoculum is a mid-range infectious dose compared to articles where influenza A virus was used to inoculated pregnant mice by intranasal route (e.g., Shi et al., 103 PFU to Chan et al 106 PFU). In a preliminary experiment, this dose of A/CA/07/2009 pandemic H1N1 did not cause clinical signs or death in pregnant female C57BL/6 mice.

  1. The specific rationale for studying transcriptomics of the fetal thymus should be better developed in the introduction.

Response:  Information supporting the rationale for studying the effects of maternal virus infections on the fetal thymic transcriptome has been added.

“In a bovine virus diarrhea virus (BVDV) maternal infection model in cattle, significant changes in gene expression in the fetal thymus and spleen were found at different time points during gestation (Smirnova 2014, Georges 2020). Moreover, gene expression was influenced by the timing of maternal infection during pregnancy (Smirnova 2014 and Knapek 2020). Clinical observations suggested that maternal BVDV infections have an impact on health of the offspring postnatally (Waldner and Kennedy 2008; Munoz-Zanzi et al 2003)”.

A few other manuscripts have evaluated the effect of influenza A virus infection on pregnancy/fetal outcomes in the mouse model some of whom were cited (first authors Littauer, Chan) and some were not (Marcelin). Please make sure that citations of other influenza studies in pregnant mice are more comprehensive as there haven’t been so many studies.

Response:  We were unable to locate a citation with G. Marcelin as author dealing with the effects of maternal influenza A virus infection of the fetal thymus.  We did find the following G. Marcelin article “Fatal Outcome of Pandemic H1N1 2009 Influenza Virus Infection Is Associated with Immunopathology and Impaired Lung Repair, Not Enhanced Viral Burden, in Pregnant Mice” which documents an increased severity of maternal disease in pregnant mice infected with influenza A virus.  This reference has been inserted into the first sentence of the second paragraph in the Introduction.  We agree that pregnant women and mice are at risk for more severe disease than non-pregnant females; however, this area of investigation differs from the focus of the present study on the effects of maternal infection on fetuses.

  1. The number of validated genes from the fetal thymus appear to be relatively few. In the fetal thymus, they validated only 3 genes. It would be ideal for the authors to validate a few additional genes in the fetal thymus as this is the main focus and innovation of the study.

Response:  This section has been edited to include 6 genes (see insert below). We validated genes of interest related to our hypothesis and validated the housekeeping genes.  Further validation of genes may be done in the future pending additional funding. We have posted all our RNA seq data in the supplementary materials so that other investigators can validate additional genes based on their specific interests.  

“Two genes in the canonical pathways Acta1 and Mylpf, two additional immune related genes Mal and Mal2, and two housekeeping genes, Hprt and Taldo1 were validated by RT-qPCR in E7.5 fetal thymuses.”

  1. Presentation of the transcriptomics analyses could be more comprehensive. The heat map shown in figure 4 that focuses only on the top 50 differentially regulated genes is not particularly helpful.  It would be more helpful to see a heat map based on canonical pathways to see how changes in gene expression between influenza cases and controls varies according to inoculation time in gestation.  It would also be nice to see heat maps that reflect specific gene sets of interest from the Broad Institute that might better reflect how the cases and controls defer for specific pathways of interest.

Response: Presentation of transcriptome analysis was expanded in Figure 4 as requested.  We agree with your assessment of the heat map in Figure 4 and it was deleted for this reason.  IPA analysis does not provide a heat map coupled with canonical pathways. However, IPA has updated their software since we completed the analysis for this paper. After re-running the analysis, we have revised Figure 4 to now include the top 10 canonical pathways in addition to the total number of genes in the pathway, the percentage of these genes that were upregulated or downregulated ³ 1.5 fold and a sliding scale of -log(pvalue). We also added panels B and C in this figure to describe the top 10 physiological system/function genes based on Adjusted P Value, and the top 10 downregulated and upregulated genes based on fold change with Adjusted P Values ³ 0.05.  We did retain the Respiratory Failure panel from the original manuscript as Panel D. 

We only completed RNA seq on day 18.5 thymus tissues after infection on day 7.5 because of budgetary constraints in the grant supporting these studies, so it is not possible at this time to compare gene expression across the days of pregnancy.  We agree that a more extensive analysis across days of pregnancy would be fantastic and hope to examine this in the future.

Regarding the Broad Institute, we would love to learn more about this possible resource.  We have studied the web site and found the mouse genome project, but have not had time to find appropriate software that might be utilized. With the short time frame in returning this manuscript back to the journal, we didn’t feel we have time to find and learn how to use this resource.  If this reviewer is willing, we would greatly appreciate a link to the resource at the Broad Institute so that we can study in context of future studies and manuscripts.

Round 2

Reviewer 1 Report

The authors fairly replied to all questions. There are inherited methodological failures—e.g., a small number of samples, selective sampling of fetuses, the lack of sex-based analysis of RNA-seq data—that cannot be fixed in the short term. The authors should consider it in future studies.

Author Response

Response to Reviewer 1

Reviewer Comment: There are inherited methodological failures—e.g., a small number of samples, selective sampling of fetuses, the lack of sex-based analysis of RNA-seq data—that cannot be fixed in the short term.

Authors' Response

We will carefully consider these appropriate suggestions in future studies.

Reviewer 2 Report

This revised manuscript is substantially improved and most of this reviewer's comments have been addressed.

Major Point 1 (Transcriptomics and whether MAL protein also affected).

The authors raise a valid point. However, I may have missed it in the revised document, but brief mention of the inability to confirm protein levels because of lack of sufficient material would be appropriate.

Major Point 2 (Reference and comparisons to related proteomic studies).

PMID:31000695

Author Response

Response to Reviewer 2

Reviewer Comment: This revised manuscript is substantially improved and most of this reviewer's comments have been addressed.

Response.  Thank you

Reviewer Comment. Major Point 1 (Transcriptomics and whether MAL protein also affected).

The authors raise a valid point. However, I may have missed it in the revised document, but brief mention of the inability to confirm protein levels because of lack of sufficient material would be appropriate.

Response.  We had to use a dissecting microscope to find and collect these fetal thymuses because they were so small.  We have inserted the following text into the manuscript to help clarify the small size of the thymuses.  The text in lines 124-127 now read as follows.

“Because of their small size, thymuses were pooled within each litter, placed in microcentrifuge tubes, frozen on dry ice and stored at -80oC for RNA extraction. The size of fetal thymuses also limited the extraction of protein from the same tissue.”

Reviewer Comment: Major Point 2 (Reference and comparisons to related proteomic studies).

PMID:31000695

Response.  Thank you for this citation.  This paper describes induction of proteins in human-induced pluripotent stem cells following 12 and 24 h in vitro culture with influenza A.  These stem cells are used as a model for human trophoblast cells.  We have added the following text to the discussion to recognize this research.

“In-vitro culture of human induced pluripotent cells with influenza A, as a model of intra-blastocyst infection, caused cytopathology, reduced viability and pluripotency, and autophagy [44] in these stem cells. Based on proteomic analysis, infection also caused dysregulation of many cellular pathways, including those directly associated with cellular differentiation.”

We can’t compare the results of our in vivo infection on E7.5 and E12.5 with fetal gene expression on 18.5 DPC to these important in vitro studies using pluripotent stem cells and proteomic approaches.  However, the research does provide very good evidence that infection with influenza very early during embryo development may have longer-term impact on placental and pregnancy outcomes.  Again, thank you for the citation. We missed this important work and hope that inclusion of the paragraph above will address this concern.
